# Combining the Classification and Pharmacophore Approaches to Understand Homogeneous Olfactory Perceptions at Peripheral Level: Focus on Two Aroma Mixtures

**DOI:** 10.3390/molecules28104028

**Published:** 2023-05-11

**Authors:** Marylène Rugard, Karine Audouze, Anne Tromelin

**Affiliations:** 1T3S, Inserm UMR S-1124, Université Paris Cité, F-75006 Paris, France; marylene.rugard@gmail.com (M.R.); karine.audouze@u-paris.fr (K.A.); 2Centre des Sciences du Goût et de l’Alimentation, CNRS, INRAE, Institut Agro, Université de Bourgogne, F-21000 Dijon, France

**Keywords:** odorants, odors, homogeneous perception, classification, UMAP, SOM, pharmacophores

## Abstract

The mechanisms involved in the homogeneous perception of odorant mixtures remain largely unknown. With the aim of enhancing knowledge about blending and masking mixture perceptions, we focused on structure-odor relationships by combining the classification and pharmacophore approaches. We built a dataset of about 5000 molecules and their related odors and reduced the multidimensional space defined by 1014 fingerprints representing the structures to a tridimensional 3D space using uniform manifold approximation and projection (UMAP). The self-organizing map (SOM) classification was then performed using the 3D coordinates in the UMAP space that defined specific clusters. We explored the allocating in these clusters of the components of two aroma mixtures: a blended mixture (red cordial (RC) mixture, 6 molecules) and a masking binary mixture (isoamyl acetate/whiskey-lactone [IA/WL]). Focusing on clusters containing the components of the mixtures, we looked at the odor notes carried by the molecules belonging to these clusters and also at their structural features by pharmacophore modeling (PHASE). The obtained pharmacophore models suggest that WL and IA could have a common binding site(s) at the peripheral level, but that would be excluded for the components of RC. In vitro experiments will soon be carried out to assess these hypotheses.

## 1. Introduction

The perception of odors involves several levels in a complex process that begins at peripheral receptors and ends in the brain [1]. In the first step, odor molecules bind to odor receptors, leading to their activation [2]. This step is governed by the combinatorial code, implying that an odorant can activate several olfactory receptors (ORs) and that an olfactory receptor can be activated by several different odorants [3,4]. Deciphering this olfactory coding remains a challenge, as most ORs are still orphans [5,6]. Moreover, the perceived odors commonly result from mixtures of odorants [7].

The perception of a mixture of odorants can be heterogeneous or homogeneous [8,9]. The percept induced by the mixture is heterogeneous when the odors of at least some of the components can be distinguished. Conversely, a mixture is described as homogeneous when a single odor is perceived from the mixture. A homogeneous perception may be of two different types [7]:An odor blending, which results from a configural processing of the mixture and occurs when the perceived odor is a new odor that differs from those of the odorants in the mixture;A complete overshadowing (or masking) when the odor of only one of the components of the mixture is recognized.

Our aim was to improve knowledge about blending and masking mixture perceptions by highlighting links between the odors and the structure of the odorants.

Establishing the relationships between odors and molecular structures remains challenging [10]. The recent studies aiming to link the odors to the structure of the odorants involve large databases and the use of computational methods applying machine learning approaches [11,12,13,14,15,16,17,18]. These approaches are applied to perform classifications of the odorants considering both their odor notes and structural features.

Another way is the pharmacophore approach, which is a ligand-based approach focusing on the necessary characteristics of the odorants to activate a potential OR target. Indeed, a pharmacophore is defined as “a set of structural features in a molecule that is recognized at a receptor site and is responsible for that molecule’s biological activity” [19,20]. The pharmacophore approach has been demonstrated to be a very effective approach [21,22], especially relevant when the receptor and consequently the structure of ligand–receptor complex are unknown, as is up to now the case with ORs [23].

With the purpose to reveal links between the odors and the structure of the odorants in the context of blending and masking mixture perceptions, we developed a classification approach similar to those previously used [14] to group the odorants of the large dataset into clusters, and then used subsets of molecules belonging to these clusters to generate pharmacophores models.

We focused on seven molecules that are components of two mixtures of odorants, a blending mixture (6 molecules) and a binary masking mixture [24,25,26,27], and also on clusters containing them.

The SOM classification based on the coordinates of the molecules in the 3D-UMAP space allowed the distribution of the seven molecular components of the two mixtures across six clusters. The odor notes of the seven components of the mixtures appeared in agreement with the odor profile of each cluster to which they belonged. Likewise, the pharmacophores generated from odorants sharing similar odors inside the same cluster revealed common features, showing good consistency between the molecular structures inside each studied cluster. The characteristics of these models suggest that WL and IA could have a common binding site(s) at the peripheral level, but that would be excluded for the components of RC.

## 2. Results

### 2.1. Overview of the Dataset

In a previous study, we built a large dataset that lists 5665 odorant molecules with their associated odors; most of these odorants are described by 2–5 odor notes [14].

Inside this dataset, we focused on seven molecules that are components of two mixtures of odorants that were previously studied at the CSGA (Table 1).

The first mixture constitutes a blending mixture called red cordial (RC mixture) and is made up of isoamyl acetate (IA), vanillin (V), frambinone (F), ethyl acetate (EA), beta-damascenone (bD) and beta-ionone (bI) [25,27]. The RC mixture is perceived as having a grenadine syrup odor. The experimental sensory tests have shown that each molecule has a distinct odor quality. Nevertheless, the odor of the mixture of these six components in specific proportions, i.e., 41.8, 41.8, 5.0, 4.3, 4.3 and 2.8% for V, F, IA, B, EA and D, respectively, is judged to be similar to those of the grenadine syrup [25]. It has been established that the presence of isoamyl acetate, vanillin, and, to a lesser degree, frambinone in the RC mixture is essential for the perception of this grenadine syrup smell [27]. In addition, the presence of the other three molecules (EA, bD, bI) typically improves the grenadine syrup without being essential.

The second mixture is a masking mixture composed of isoamyl acetate (IA) and whiskey lactone (WL) [24]. This mixture, which is noted as WL/IA, constitutes a masking in which a qualitative dominance of the woody note of WL occurs when the perceived intensities of unmixed WL and IA are equal.

### 2.2. Dimensions Reduction and Clustering

In a previous study [14], we had performed a dimension reduction in a two-dimensional space by various methods, of which the uniform manifold approximation and projection (UMAP) was used here [28]. We had established that the combination of UMAP and the k-means classification has been the most relevant technique for discriminating structure-odor relationships.

#### 2.2.1. Dimensions Reduction

The molecular structures of the 5665 molecules were encoded by 1024 fingerprints, and reduced to three-dimensional data using the UMAP reduction method.

We observed that the elements were split in the 3D-space into three main areas, i.e., Aa, Ab, Ac and Ad (Figure 1).

#### 2.2.2. Clustering

The previous clustering calculations were performed from coordinate values in 2D [14]. In the present work, we used the coordinate values of the molecules in the 3D-UMAP space to make and compare the clusters obtained using two classification algorithms, k-means and SOM.

We used the “Elbow” curve to determine the optimal number of clusters and, in addition, the Kelley penalty score to determine the optimal number of clusters. For each UMAP dimensional space, both the minimum score and the optimal number of clusters are 4 (Appendix A). Nevertheless, we selected level 9 and 16 clusters to refine the analysis via k-means and SOM with 4, 9 and 16 clusters. We called the classification into 4, 9, and 16 clusters “classification at level L4”, “classification at level L9” and “classification at level L16”, respectively.

The clusters obtained were called depending on the used classification method (k-means or SOM), then by the classification level, followed by the cluster number. For example, cluster 1 of the SOM classification with 4 clusters is named “SOM4-Cl-1”. A similar notation is used for clusters at levels L9 and L16.

The allotments of the elements across the clusters at levels L4, L9 and L16 are displayed in Figure 2. The SOM classifications at levels L9 (a grid of 3 by 3) and L16 (a grid of 4 by 4) provide empty classes (SOM9-Cl-5, SOM9-Cl-6, SOM16-Cl-7, SOM16-Cl-9 and SOM16-Cl-12). Thus, there are only 7 clusters SOM9 and 13 clusters SOM16.

In line with the aim of our study, we focused on the allotments among the clusters k-means and SOM of the molecules involved in the RC mixture and the masking mixture WL/IA. The location of these molecules in the areas and in the clusters is reported in Table 2.

The area Aa encompasses two molecules from the RC mixture, bD and bI, and WL (masking mixture WL/IA). At each level of the clustering, bI and bD are gathered in the same cluster. Except at level L4, WL belongs to a cluster separate from that of bI and bD.

Both V and F belong to the area Ab, also to the same k-means cluster at all levels, and to the same SOM cluster at levels L4 and L9. Conversely, the two molecules are placed in two separate SOM clusters at level L16 (SOM16-Cl-2 and SOM16-Cl-5).

The two esters IA and EA are in the area Ac. As observed in the case of V and F, these two molecules are gathered in the same cluster k-means at all levels of clustering, and in the same cluster SOM at levels L4 and L9. Conversely, these two molecules belong, respectively, to the SOM clusters SOM16-Cl-13 and SOM16-Cl-14.

Using the k-means16 classification, several odorants were determined to belong to the same cluster, including the following: bD and bI (k-means16-Cl-7), V and F (k-means16-Cl-8) and IA and EA (k-means16-Cl-4). Conversely, except for bD and bI, the SOM16 classification places each component of the mixture in a separate cluster. Because of this better distribution that split the components into separate clusters, we, therefore, selected the clusters SOM16 that contained the components of the mixture in the following.

A detailed description of the clusters is reported in Appendix B.

### 2.3. Distribution of Odor Notes within Clusters

The number of occurrences of the odor notes varies in a large range from 5 (tallow) to 1790 (fruity). Consequently, the direct comparison of the number of occurrences of odor notes in each cluster is irrelevant. Hence, we considered the following relative amounts related to each cluster [14]:the relative frequency of every odor notes compared to its frequency in the database (Equation (1));the relative frequency of odorants carrying each odor note compared to the number of molecules in the considered cluster (Equation (2)).
(1)$$\%\;odor\;note=\%ON=\frac{\mathrm{Number}\;\mathrm{of}\;\mathrm{occurrences}\;\mathrm{of}\;\mathrm{an}\;\mathrm{odor}\;\mathrm{note}\;\mathrm{in}\;\mathrm{the}\;\mathrm{cluster}}{\mathrm{Total}\;\mathrm{number}\;\mathrm{of}\;\mathrm{occurrences}\;\mathrm{of}\;\mathrm{this}\;\mathrm{odor}}$$
(2)$$\%\;odorant\;molecule=\%OM=\frac{\mathrm{Number}\;\mathrm{of}\;\mathrm{occurrences}\;\mathrm{of}\;\mathrm{an}\;\mathrm{odor}\;\mathrm{in}\;\mathrm{the}\;\mathrm{cluster}}{\mathrm{Number}\;\mathrm{of}\;\mathrm{elements}\;\left(\mathrm{molecules}\right)\;\mathrm{in}\;\mathrm{this}\;\mathrm{cluster}}$$

The values of the number of occurrences of the odor notes in the whole database and per cluster are reported in Appendix A.

We examined the occurrences of the 25 most frequent odor notes across the 6 clusters SOM16. The radar plots of the %ON odor profiles are displayed in Figure 3 (Additionally, the %OM odor profiles across the clusters are displayed in Appendix A).

Furthermore, the odor description of the components of mixtures implicates 39 odor notes that have at least 5 occurrences (Table 1). Some of these odor notes are common to the list of the 25 frequent odor notes, but several other less frequent notes are characteristic of the odorant profile for the molecules related to a specific cluster. Thus, we examined the distribution of these 39 odor notes, and their radar plots of the %ON odor profiles are displayed in Figure 4.

Vanilla, as well as spicy and balsamic odor notes, characterize the cluster SOM16-Cl-2, to which vanillin belongs. Indeed, nearly 80% of vanilla notes and about 25% of spicy and balsamic notes are gathered in this cluster (Figure 3a and Figure 4a). Odorants carrying the spicy, balsamic and vanilla notes constitute 18%, 12% and 11% of SOM16-Cl-2, respectively (Appendix A). The odorant description of vanillin also involves chocolate and creamy notes; the cluster SOM16-Cl-2 encompasses, respectively, 16% and 24% of the occurrences of these two odor notes. Additionally, more than 40% of the occurrences of powdery and grape notes are clustered in this cluster.

The cluster SOM16-Cl-4, which contains WL, encompasses nearly 40% of caramellic notes and more than 60% of the molecules carrying the odor notes such as coconut (67%), coumarinic (64%), lactonic (80%), and, to a lesser extent, celery (50%) and tonka (60%) notes (Figure 3b and Figure 4b).

Frambinone belongs to SOM16-Cl-5, which is dominated by the floral, balsamic and rose notes, despite less than 25% of these three odor notes being gathered in this cluster (Figure 3c). Nevertheless, the “floral” molecules constitute nearly 50% of this cluster (Appendix A). In addition to the floral notes, the odor profile based on the 39 odor notes indicates the presence of the odor notes plum (23%) and raspberry (19%) (Figure 4c).

More than 30% of the molecules carrying the raspberry note are gathered in the cluster SOM16-Cl-8, which also contains two molecules of the RC mixture, bI and bD. Nevertheless, the odor profile of SOM16-Cl-8 is different from that of SOM16-Cl-5 because the tobacco, violet and orris notes are especially frequent in SOM16-Cl8 (43, 44 and 48%, respectively) (Figure 4d).

The esters IA and EA belong, respectively, to clusters SOM16-Cl-13 and SOM16-Cl-14, which are characterized by the frequency of the fruity note. Indeed, fruity odorants constitute 65% of SOM16-Cl-13 and 55% of SOM16-Cl-14 (Appendix A). Both clusters gather more than 20% of apple, tropical and pineapple odorants; in other words, the whole of the two clusters contains 60, 49 and 75% of apple, tropical and pineapple molecules, respectively (Figure 3e,f).

Considering the most frequent odor notes, the difference between SOM16-Cl-13 and SOM16-Cl-14 is due to the various frequencies of several odor notes. First, at least 30% of the occurrences of fatty, waxy and oily molecules are gathered in SOM16-Cl-13, compared to approximately 10% in SOM16-Cl-14. Furthermore, it appears that the winey, pear and brandy odor notes are characteristic of the odorant profile of SOM16-Cl-13 (46, 54, and 58% of the odorant carrying winey, pear and brandy notes, respectively, belong to SOM16-Cl-13); a somewhat greater portion of banana molecules (%ON = 43%) belong to SOM16-Cl-13 than to SOM16-Cl-14 (30%). In addition, sharp is unique to SOM16-Cl-14; 38% of the occurrences of this odor are noted in SOM16-Cl-14, compared to less than 10% in SOM16-Cl-13. Similarly, approximately 20% of odorants that have caramellic, berry and creamy notes belong to SOM16-Cl-14, while less than 10% of such odorants are in SOM16-Cl-14.

### 2.4. Co-Occurrences of Odor Notes across the Clusters SOM16

In addition to the odorant profile examinations, we look at the co-occurrences of the odor notes (Appendix A). We focused on 55 odor notes (the 25 most frequent odor notes in addition to the 39 odor notes of the mixtures, 10 common to both). We used the nonsymmetrical square matrix of odor notes obtained from the co-occurrence symmetric square matrix by weighting the number of associations by the frequency of occurrences to generate the heatmaps displayed in Figure 5.

Vanillin belongs to the cluster SOM16-Cl-2, where the note vanilla is associated with the notes sweet, chocolate and creamy. For the entire database, vanilla is mainly associated with a sweet note (63% of its occurrences) and with creamy, spicy (both 25%), floral (18%), woody and balsamic (both 11%) odor notes. In the odorant descriptions of the compounds that belong to SOM16-Cl-2, the frequencies of associations are quite similar but occur slightly more frequently with creamy (27%), spicy (31%) and balsamic (19%) notes. The occurrences of vanilla to chocolate is approximately 10% in the entire base as well as in SOM-Cl-2. Nevertheless, the occurrences of chocolate notes are noticeably more frequently associated with vanilla, creamy and sweet notes in SOM-Cl-2 than in the entire base. Indeed, 70% of the occurrences of chocolate notes are associated with sweet and 60% with both vanilla and creamy notes, while those are 38%, 21% and 19%, respectively, in the entire base.

The cluster SOM16-Cl-4 contains WL and is characterized by caramellic, lactonic, coumarinic, coconut, tonka and celery odor notes. There are several co-occurrences between these odor notes, especially between coconut and coumarinic, lactonic, tonka and toasted (42%, 31%, 42% 21% of the occurrences of these four odor notes in the entire database and 59%, 33% and 60% in SOM16-Cl4, respectively). Conversely, there are few associations between these six odor notes and the celery, caramellic, woody, nutty, burnt, and toasted notes, which themselves have few co-occurrences in the database except for toasted and nutty (43% of occurrences of toasted notes). However, only 20% of the occurrences of the toasted note are associated with the nutty note in SOM16-Cl-4. In contrast, caramellic and burnt notes are associated more frequently in SOM16-Cl-4 than in the entire base (50% against 19% of the occurrences of the burnt note).

In SOM16-Cl-5, we considered the following odor notes: fruity, sweet, floral, rose, balsamic, berry, plum, raspberry and ripe. The fruity, sweet and floral notes are strongly associated among themselves, as well in the database as in SOM16-Cl-5. The rose, balsamic, berry, plum, raspberry and ripe notes are also strongly associated with fruity, sweet and floral notes. Although only 15% of occurrences of ripe notes are associated with floral notes in the base, the unique occurrence of this odor note in SOM16-Cl-5 is associated with fruity, sweet and floral. Conversely, there are relatively few co-occurrences between rose, balsamic, berry, plum and raspberry notes, as well as fruity, sweet, and floral notes, among themselves. Balsamic, berry and raspberry notes are more frequently associated with the rose note in SOM16-Cl-5 than in the entire base (25% against less than 15%); plum and rose notes have no co-occurrence in SOM16-Cl-5. Moreover, 25% of the occurrence of the raspberry note is associated with berry and plum notes (against 14% and 7% in the entire base, respectively).

The odorant descriptions of bI and bD involve 16 odor notes, to which the minty and herbal odor notes that are specific to SOM16-Cl-8 must be added, for a total of 18 odor notes. The herbal and minty notes are well associated with each other in the entire base, as in SOM16-Cl-8 and slightly more in SOM16-Cl-8. These two notes are also associated with fruity, sweet and floral notes (at least 20% of their occurrences except for the minty and floral notes; 11% and 12% of the occurrences of minty occur in the entire base and in SOM16-Cl-8, respectively). Conversely, the co-occurrences of minty and herbal notes with the odor notes of bI and bd do not exceed 5%, with the exception of the association between herbal and woody, which reaches 37% of the occurrences of herbal in SOM16-Cl-8 (19% in the entire base). There are few associations among the odor notes that are involved in the descriptions of both bI and bD in the entire base, but they are more frequent in SOM16-Cl-8 except for the woody note, which is well associated with most notes. The wrong association is between woody and apple notes in the entire base (4% of the occurrences of apple) but reaches 23% in SOM16-Cl-8. Finally, the woody note is the most frequent odor note associated with raspberry, especially in SOM16-Cl-8 (74% of the occurrences of raspberry against 40% in the entire base).

We focused on the most frequent 15 odor notes involved in the cluster SOM16-Cl-13 by embracing the most frequent notes in the entire base, as well as those involved in the mixtures. All these odor notes are strongly associated with fruity and in a minor part with green and sweet notes, as well as in the entire base as in SOM16-Cl-13. Nevertheless, there is a little more relative frequency of the association with fruity in SOM16-Cl-13 as in the entire base, but somewhat less with the green note and much less with the sweet note. This is especially the case for pear and banana notes, and these two odor notes are more closely associated among themselves in SOM16-Cl-13 than in the entire base. Moreover, the solvent note is approximately four times more associated with pear and banana notes, while all occurrences of solvent and ripe are associated with the fruity note in SOM16-Cl-13, compared to 38% and 81% in the entire base, respectively. The occurrences of fruity, green and sweet notes are more associated with winey and brandy notes in SOM16-Cl-13. Winey, pear, banana and brandy notes are slightly more associated among themselves in SOM16-Cl-13, except for winey and banana notes (for them, there is almost no difference between association) in the entire base and SOM16-Cl-13).

By combining the most frequent notes belonging to the molecules of SOM16-Cl-14, we retained 13 odor notes, of which 7 were related to EA (fruity, sweet, weedy, green, sharp, brandy and winey). As observed for the odor notes of SOM16-Cl-13, all notes are strongly connected to fruity, and the relative co-occurrences are approximately the same for the entire base and for SOM16-Cl-13, with the exception of the sulfurous note, which is two times more associated with fruity in SOM16-Cl-13 (%ON = 41% in SOM16-Cl-13 against 19% in the entire base). The associations with green and sweet notes are also less frequent with the association with fruity and global notes in SOM16-Cl-14 than in the entire base except for brandy (respectively three and two times more frequent with green and sweet in SOM16-Cl-14). Similarly, the relative co-occurrences of ethereal with pineapple, winey, sharp and weedy are two times more frequent in SOM16-Cl-14. Brandy has only two occurrences in SOM16-Cl-14, and only one with winey, sharp and weedy notes, which correspond to the odorant description of EA.

### 2.5. Selection of Subsets of Odorants Based on Odor Profiles

Within each of these clusters, we selected approximately 10 molecules having an odor profile close to those of the components of the mixtures. For that, within each cluster, for each molecule, the number of common odor notes, the number of noncommon odor notes, the percentage of common odor notes and the percentage of noncommon odor notes with the molecule of the mixture were calculated.

We selected all the molecules that had at least three common odor notes and no more than two uncommon odor notes. We refined the selection according to the following criteria:If in these first criteria, more than 10 molecules were selected, we again selected these molecules in which the percentage of noncommon odor notes was the smallest;If there was still a need to restrict the number of molecules selected, we selected the molecules that had the highest percentage of common odor notes.

The list of molecules that make up each subset is displayed in Table 3.

### 2.6. Pharmacophore Study

We used the PHASE module [29] to set the chemical features of the odorants and identify the critical common features present in a set of odorants. We developed pharmacophores in the following ways:From groups based on the mixture components and containing at least two molecules of the mixtures;From each molecule component of the mixtures: each molecule is, in fact, an ensemble of its conformers (energy range of 21 kJ/mol) and was denoted “M-c”, where M is the molecule, and c symbolizes conformers;From the subsets of molecules selected in the SOM16 clusters: the subsets of various molecules were named by the initial of the referred molecule followed by the initial “s” (for subset). For example, the vanillin subset selected on the basis of the odorant profile was named “V-s”.

Every pharmacophore was developed following a ligand-based model with at least three features, and all molecules were considered active molecules. For all hypotheses, the aromatic rings were considered as hydrophobic groups. Knowing that a hydrogen bond donor is also an acceptor, we used only the hydrogen bond acceptor (A) feature. The PhaseHypoScore ranks the hypotheses from the worst to the best, allowing us to identify the most significant hypothesis.

In the following text, the hypotheses relating to each molecule are named by the abbreviation “hyp” (for pharmacophore hypothesis) followed by the initial name of the molecule. For example, the hypothesis developed from the vanillin V-s subset was named “hyp-V-s”.

#### 2.6.1. Pharmacophores Generated from Subsets Composed of Components of the Mixtures

Pharmacophores generated from the subsets based on the “Red Cordial” mixture

We considered the set of the six molecules and several subsets involving at least two molecules of the RC mixture. Only one hypothesis could be created from the V-IA-F-EA-bD-bI-s subset (Figure 6a). The partial hit for this hypothesis concerns only IA, bD and bI, so it does not match all active molecules. The details of the pharmacophore are presented in Appendix A. 

According to a previous study [27], the presence of V, IA and F has a significant impact on the perception of the blending mixture. However, compared to the other two molecules, F is less involved in the perception of grenadine syrup odor. Therefore, we focused on V, IA and F to generate pharmacophore hypotheses from three subsets encompassing at least V and one of the two other main components of the RC mixture, the V-IA-F-s, V-IA-s and V-F-s subsets.

Three pharmacophore hypotheses were generated from the V-IA-F-s subset. The mapping of the molecules is displayed in Figure 6b–d.

The two first pharmacophores (AAR_1 and AAR_2) possess an aromatic ring and hydrogen bond acceptors and match V and F but not IA. The last hypothesis, AAH_1, which maps V and IA, contains two hydrogen bond acceptors and a hydrophobic feature that corresponds to the aromatic cycle of V. The poor score values indicate the weak significance of the hypotheses.

The pharmacophore generations carried out from the V-IA-s and V-F-s subsets also provided poor results. The single hypothesis obtained from the V-IA-s subset is identical to hypothesis AAH_1 generated from the V-IA-F-s subset. Likewise, the two hypotheses generated from the V-F-s subset are identical to those generated from the V-IA-F-s subset (AAR_1, AAR_2).

2.Pharmacophores generated from the subset WL/IA-s

Nine pharmacophores were generated from the two molecules, and the first three were characterized by good PhaseHypoScore values (Appendix A). The three best hypotheses and the mapping of the molecules are displayed in Figure 6e–g.

These hypotheses consist of the same features, two A and one H, but differ in their geometries, as indicated by the distances and angle values. Indeed, the AAH_2 hypothesis has a linear geometry (angle $\hat{A1A2H4}$) 149.7 deg) similar to those of hypotheses AAH_1 and AAH_3 (angle $\hat{A1A2H4}$ approximately 100 deg). The distances A1-A2 vary from 2.25 Å (AAH_1 and AAH_2 hypotheses) to 2.27 Å (AAH_3 hypothesis). The distances A1-H4 are respectively 2.58 and 3,56 Å for AAH_1 and AAH_2 hypotheses; the distance A1-H3 is 3.83 Å for AAH_3 hypothesis.

#### 2.6.2. Pharmacophore Generated from the Conformers of Each Single Molecule Component of the RC Mixture and WL/IA Masking

In this case, each molecule is considered an ensemble of its conformers, and the obtained hypothesis represents the pharmacophore of its conformational space. The pharmacophore hypothesis generated from each molecule of the two mixtures is displayed in Figure 7.

All hypotheses include at least one hydrogen bond acceptor feature (A) and one hydrophobic feature (H), with the exception of hyp-F-c, which does not contain hydrophobic features but an aromatic ring (R) (Figure 7c). Conversely, the hyp-V-c hypothesis contains both R and H features (Figure 7a). The hypotheses hyp-IA-c (Figure 7b), hyp-EA-c (Figure 7D), hyp-bD-c (Figure 7e), hyp-bI-c (Figure 7f) and hyp-WL-c (Figure 7g) encompass various numbers of H and A features: from two A and one H for hyp-EA-c to one A and five H for hyp-bD-c; hyp-bD-c and hyp-bI-c are richer in hydrophobic features.

All the inter-feature distance values are reported in Appendix A. By comparing the distances between the A and H features for the hypothesis of the molecular components of RC mixtures, very few close values between the A and H or R features were observed. Indeed, there are three distances A-H for hyp-V-c (1.424 Å, 4.165 Å, 5.167 Å), and only one of them matches those of hyp-IA-c (A1-H3, 4.287 Å). Conversely, the distances A3-R6 (2.767 Å) and R6-A1 (2.796) of hyp-V-c are close to the A2-R4 distance of hyp-F-c (2.758 Å), while R4-A1 of hyp-F-c (5.967 Å) is close to A2-H3 of hyp-IA-c (5.815 Å). However, the distances between the two hydrophobic features, H or R, vary greatly according to the hypotheses (R6-H5, 3.734 Å for hyp-V-c; H4-H3 2.468 Å for hyp-IA-c). Moreover, despite quite similar structures regarding the number of hydrophobic features between hyp-bD-c and hyp-bI-c, there are no common A-H distances (ranging from 3.225 to 3.853 Å and from 5.082 to 5.803 Å, respectively, for hyp-bD-c and hyp-bI-c).

In contrast, there are several close values regarding the A-H and H-H distances of the hyp-IA-c and hyp-WL-c hypotheses, including the following: 1.828 Å, 3.558 Å, and 4.287 Å for the A1-H4, A2-H4 and A1-H3 distances of hyp-IA-c, respectively, and 1.781 Å, 3.686 Å, 4.273 Å for the A1-H5, H3-A1 and A1-H4 distances of hyp-WL-c, respectively. Moreover, the H-H distances are also close (2.468 Å for hyp-IA-c and 2.361 for hyp-WL-c). A similar observation can be made for A-A distances of 2.303 Å and 2.255 Å for hyp-IA and hyp-WL, respectively.

#### 2.6.3. Pharmacophores from the Subsets of Molecules Having Similar Odor Profiles

We developed pharmacophore hypotheses from each subset resulting from the selection of the molecules in the clusters SOM16 on the basis of their odor profiles (Table 3): V-s (12 molecules), IA-s (11 molecules), F-s (10 molecules), EA-s (11 molecules), bD-s (7 molecules), bI-s (10 molecules) and WL-s (9 molecules). The details of the PHASE-generated top hypotheses are reported in Appendix A and the interfeature distances in Appendix A.

The best hypotheses obtained from each subset and the mapping of the related molecules are shown in Figure 8. The structures of the molecules align in an ordered way on each pharmacophore hypothesis as follows: the rings of each molecule are aligned on each other, the carbon chains are in close spaces and the oxygen atoms map the A features, although in a lesser ordered arrangement in the cases of bD (Figure 8e) and bI (Figure 8f). All molecules in each subset map the related best significant hypotheses with the exception of frambinone, which does not map the hyp-F-s model.

The best hypotheses models that are related to RC mixtures were diversely made. All encompass at least one hydrogen bond acceptor A, except hyp-bD-s, which contains only H. Hyp-V-s encompasses the three features A, H and R, and hyp-bI-s possesses two H. Four hypotheses, hyp-IA-s, hyp-EA-s, hyp-WL-s and hyp-F-s, have two hydrogen bond acceptors, while hyp-F-s does not contain hydrophobic feature H but an aromatic feature R.

#### 2.6.4. Pharmacophore Comparisons

To evaluate the similarities between the hypotheses, we compared the three-dimensional arrangement of the features of these hypotheses by aligning them using the “Hypothesis Alignment” task, which aligns hypotheses in pairs using one of the two hypotheses as a template. The root-mean-squared deviation (RMSD) between the features of the reference hypothesis and the second hypothesis allows us to evaluate the quality of the alignment of the structures (Table 4).

We focused on the three main components of the RC mixture (V, IA and F) and WL. The comparisons in pairs were performed between:The hypotheses generated from the molecules: hyp-V-c, hyp-IA-c, F, and hyp-WL-c;The hypotheses generated from the subsets of molecules having odor profiles similar to those of components of the mixtures: hyp-V-s, hyp-IA-s, hyp-F-s, and hyp-WL-s;The hypothesis of each molecule with the hypothesis generated from the subset of similar odor profiles.

By the pairwise comparison between the hypotheses generated from the molecules, a mediocre overlap was obtained for the hyp-V-c and hyp-IA-c hypotheses (Figure 9a), corresponding to a poor RMSD value. No reliable result was obtained for hyp-V-c and hyp-F-c alignment (Figure 9 b) or hyp-IA-c and hyp-F-c alignment (Figure 9c). Conversely, the hyp-IA-c and hyp-WL-c hypotheses were quite satisfactorily aligned (Figure 9d), as reflected by a good RMSD value.

The comparison between the hypotheses generated from subsets V-s, IA-s, and F-s, show inadequate overlap between the hydrogen bond acceptors and the hydrophobic features. Indeed, there are very close positions for the following:A and R (hyp-V-s and hyp-IA-s comparison (Figure 10a);A and H (hyp-V-s and hyp-F-s comparison, Figure 10b);A and R (hyp-IA-s and hyp-F-s comparison, Figure 10c).

Such mapping would suggest a colocalization of a hydrogen bond donor and a hydrophobic site at the binding site, which is not rational.

Conversely, the comparison between the hyp-WL-s and hyp-IA-s hypotheses provided a satisfactory mapping due to a good overlap of their respective A and H features (Figure 10d).

The mappings obtained by comparison for the pairs hyp-V-c/hyp-V-s and hyp-IA-c/hypIA-s, and to a lesser extent hyp-WL-c/ hyp-WL-s, appear satisfactory. Indeed, there is a substantial overlap for the pairs of features R7-R8, H5-H6, and A1-A4 for hyp-V-c/hyp-V-s (Figure 11a). Similarly, the alignment of hyp-IA-c and hyp-IA-s shows good mapping between H6 and H7, A1 and A3, and A2 and A4 (Figure 11b). However, due to the alignment of hyp-WL-c and hyp-WL-s, the overlays between A and H features exclude one H of each model (Figure 11d). Conversely, no reliable alignment could be achieved for F-s/F-c (Figure 11c, no RMSD value).

## 3. Discussion

The present work aimed to qualitatively highlight the links between the structural properties of odorants and their perceived odor to improve the understanding of the homogeneous perception of aroma mixtures. For that purpose, we focused on seven molecules belonging to a large database containing 5665 odorants.

Previously, we used this database to highlight the links between the molecular structures and the odor notes using several dimension reduction methods associated with clustering approaches [14]. Because we have pointed out the advantages of the UMAP dimensional reduction method, we chose this method and extended the distribution of the elements in a 3D space. We evaluated two clustering methods, k-means and SOM, using the 3D coordinates in the UMAP space. We selected the SOM method using a grid of 4 by 4 for its effectiveness in discriminating between the odor profiles of the 13 clusters thus obtained (Figure 3 and Figure 4).

The examination of the odor notes showed a good agreement between the odor profiles of the components of the mixtures and those of the SOM clusters to which they belong; this is not always true for the k-means clusters. This difference between the results provided by the two clustering methods is especially pointed out in the cases of V and F on the one hand and IA and EA on the other hand.

V and F are gathered in the cluster k-means-Cl-8. Nevertheless, despite the similarity of the chemical structure due to the phenol moiety, V and F have very different odor profiles (vanilla-chocolate vs. fruity-raspberry). Thus, the specific odorant profiles of V and F are in better agreement with the distribution of these molecules in the separate clusters SOM16-Cl-2 (Figure 4a) and SOM16-Cl-5 (Figure 4c).

As esters, IA and EA are located in the same area of the UMAP space, and both belong to k-means-Cl-4. These two molecules share a fruity odor but differ in several other notes: IA is described by banana-pear notes while EA is ethereal-green-sharp. The analysis of the relative frequencies (% ON) of the odorants shows that fatty, waxy, oily, winey, pear and brandy molecules are largely more frequent in SOM16-Cl-13 where IA is located, whereas ethereal and sharp notes more specifically characterize SOM16-Cl-14 where EA is located (Figure 3e,f and Figure 4e,f and Appendix A).

These examples underline the efficiency of the SOM clustering method to satisfactorily discriminate the odor profiles related to some specific molecular features of the odorants. Moreover, the benefit of using SOM clustering concerns the number of effective clusters. At the same dividing levels L9 and L16, the number of SOM clusters decreased, respectively, to 7 and 13 (Figure 2). This could be due to the flexibility of the SOM algorithm, which can better adapt the number of clusters to the topology of the data than k-means. That does not mean that SOM decisively outperforms k-means; several groups of odorants defined by k-means or overlaps between k-means and SOM clusters could be useful to other analyses in future studies.

The analysis of the co-occurrences especially highlights the specificity of the association between odor notes according to the clusters (Figure 5). Moreover, the examination of several associations between odor notes suggests some ways to refine the understanding of structure-odor relationships. For example, the raspberry note is common to F, which belongs to cluster SOM16-Cl-5; bI and bD are gathered in cluster SOM16-Cl-8. The examination of the co-occurrences of the raspberry note in the clusters SOM16-Cl-5 and SOM16-Cl-8 points out privileged associations between raspberry and floral in SOM16-Cl-5 and between raspberry and woody in SOM16-Cl-8. Thus, these two types of raspberry odorants would be related to different types of chemical structures. In the case of F, this suggests little raspberry typicity and the dominance of the floral note, in accordance with the description reported by Arctander that indicates “*Very sweet, fruity, warm odor resembling Raspberry preserves. Sweet-fruity taste but not very powerful*” [30].

Both bD and bI belong to SOM16-Cl-8, which is characterized by woody and minty notes (Figure 3e and Figure 4e). The odorants bD and bI have one of the most complex descriptions, involving respectively 11 and 8 odor notes (four notes in common: fruity, sweet, floral, raspberry; rose, apple, tobacco, natural, grape, plum, and tea are specific to bD; woody, tropical, berry, dry, powdery, violet and orris are specific to bI). There are few, or not many, co-occurrences between minty and the odor notes involved in the odorant descriptions of bI and bD that suggest that another specific group of odorants characterized by the minty note is embedded in SOM16-Cl-8 but is distinct from the odorants carrying floral notes such as violet, orris, and rose.

To assess the specific structural characteristics of the molecules involved in the two mixtures, we performed a pharmacophore study using a threefold approach for purposes of comparison between the generated hypotheses. In addition to the generation of pharmacophores using single or several molecules of the mixtures, we performed pharmacophore generation using subsets of a dozen selected molecules in each of the six clusters on the basis of their odorant descriptions.

The results provided by the pharmacophore approach revealed that the RC mixture and the masking mixture WL/IA make up two different cases.

Despite WL and IA belonging to different structural groups, the results involving these two molecules indicate the existence of some common molecular features. Indeed, significant hypotheses were obtained from the subsets: WL/IA-s (Figure 6e–g) and WL-s and IA-s based on odorant profiles (Figure 8b,g). In all cases, there is good overlapping as well as good alignment of the molecules with the features of the models. Moreover, the results obtained by pairwise comparisons of pharmacophores put forward a similarity between the geometry of the hypotheses generated as well from single molecules of WL-c and IA-c (Figure 9d) as from the subsets WL-s and IA-s (Figure 10d). All these statements suggest that WL and IA would share common or similar binding mode(s) and/or binding site(s) at the peripheral level of the olfactory system.

In contrast, no common feature emerged from the pharmacophores generated using all six or only three or two of the components in the RC mixture (V-IA-F-s, V-IA-s and V-F-s subsets). Moreover, no satisfactory overlapping was obtained by in pair comparisons of pharmacophores hyp-V-c, hyp-IA-c and hyp-F-c generated from the single molecule subsets on the one hand (Figure 9a–c), and hyp-V-s, hyp-IA-s and hyp-F-s generated from subsets V-s, IA-s, and F-s on the other hand (Figure 10a–c). That means that each key component of the RC mixture has a specific spatial structure that differs from that of the other components. It is especially interesting in the case of V and F, which, despite their common phenolic group, belong to distinct SOM16 clusters and have distinct odor profiles without a common odor note. Consequently, the key components of the RC mixture do not share a common binding site or binding mode.

Examination of the hypothesis models generated from the subsets based on odorant profiles presents some suitable alignments of the molecules with the features of the models (Figure 8). There are also satisfactory pharmacophore mappings by pairwise comparison between pharmacophore generated from the single molecules and from the subsets based on odorant profiles (Figure 11). That means that each component of the mixture shares common features with numerous molecules belonging to the same cluster. That is true for V, IA and WL, except for F. Indeed, neither molecule F nor the model hyp-F-c can map the hypothesis model hyp-F-s. That could suggest less efficient binding to some ORs and be related to the weak “raspberry” odor note described in several databases [30,31].

In summary, the results provided by the obtained pharmacophore models suggest that RC mixture blending perception does not originate from competition at the same binding site at the peripheral level. Nevertheless, that does not exclude the binding at different sites of the same OR, as is, for example, the case for the broadly tuned OR1G1 [32,33,34,35,36]. Moreover, such binding of ligands at different sites on the same receptor can produce blocking by the antagonist or inverse agonist effects or synergistic effects by allosteric modulation of the olfactory signal [37,38,39,40]. In vitro functional activity studies actually performed in our research group will give an answer to this question. If no common OR is revealed by our ongoing work, that would mean that the olfactory “grenadine syrup” identity probably involves the integration of the olfactory signal at higher levels.

Conversely, the pharmacophore models generated from the components of WL/IA masking suggest that these ligands could interact at the common binding site(s), as also suggested by a previous work [26,41]. This assumption will also be asserted or discarded by the results of in vitro functional tests currently in progress. Obviously, masking occurring at the peripheral level does not exclude the processing of olfactory signals at higher levels.

More largely, our study highlights how the association of the UMAP dimensional reduction and SOM clustering methods allows for the partitioning of odorants into groups whose odor profiles reflect an interesting specificity regarding the molecular structures. It is especially revealed in the present study by the cases of vanillin and frambinone on the one hand and isoamyl acetate and ethyl acetate on the other hand, which are allocated by SOM clustering in separate clusters in adequacy with their odorant profiles despite some common molecular features (phenol moiety for V and F, ester function for IA and EA).

## 4. Materials and Methods

### 4.1. Description of the Dataset and Molecules of Interest

For this study, a dataset of 5665 smell compounds and 162 odor notes that occurred at least 5 times [42] were extracted and compiled from the databases “The Good Scents Company” [43] and “Flavor Base” [31]. All data are available upon request.

Within the database, we focused on the components of two mixtures: a blending mixture (red cordial mixture, RC mixture) and a masking mixture (noted WL/IA). The RC mixture was made of six odorants: vanillin (V), isoamyl acetate (IA), frambinone (F), ethyl acetate (EA), beta-damascenone (bD) and beta-ionone (bI) [25,27]. The WL/IA masking mixture was composed of whiskey lactone (WL) and isoamyl acetate (IA) [24].

In addition, we focused on several subsets that were defined by the classification methods and the odorant profile of the molecules. The list of the molecules of these subsets is detailed in Table 3 and Appendix A.

### 4.2. Fingerprint Generation, Dimension Reduction and Clustering

The structure of each molecule was converted with KNIME software (v 3.6.2) to extended-connectivity fingerprints (ECFP) of 1024 bits in the same way as performed in a previous study [14].

The uniform manifold approximation and projection (UMAP) dimensional reduction technique was performed using the R package umapr (version 4.1.1) [44] from the fingerprint values [28].

The k-means and SOM (Kohonen Self-Organizing Map) [45] classifications were performed and based on the coordinates in the UMAP 3D space using R software (version 4.1.1) [44]. The Kohonen package was used for the SOM calculation [46].

Both k-means and SOM are nonhierarchical clustering methods for which the desired number of clusters k must be predefined. Nevertheless, there are several differences and specificities between these two clustering algorithms [47,48,49].

The k-means algorithm is a well-known method that is among the most popular clustering techniques. Nevertheless, the k-means algorithm is sensitive to outliers and is less efficient if clusters are not hyperspheres [50]. Indeed, k-means is a vector quantization method with the limitation that the final classification depends on the initial selection of the number of centroids and seeds, while nodes (centroids) are independent of each other. With k-means clustering, the partitioning of n observations into k clusters was achieved, in which each observation belongs to the cluster with the nearest mean (cluster centroid).

SOM classification [51], which is less widely used, is a nonlinear mapping method and neural network model by which the clusters with observation models, represented by nodes, are formed geometrically, while the number of neurons in the output layer has a close relationship with the class number in the input stack. SOM maps high-dimensional data in an orderly fashion on a two-dimensional grid and can change its internal structure under the influence of elements from the same system by preserving the original topology of the dataset with the aim of optimally describing the domain of observations. SOM allows to represent the similarity of data and cluster data by placing the most similar nodes in the same class or neighboring classes closer, and the dissimilar nodes farther apart in the grid.

The optimal number of clusters was determined using both the elbow curve and the Kelley penalty score [52]; the calculations were performed respectively using the R packages map tree (kgs function) and factoextra [44].

The visualization of the distribution of the molecules and clusters in the 3D-UMAP space was performed using the package XLSTAT-3Dplot [53].

### 4.3. Construction of Pharmacophores

The pharmacophores were generated using the PHASE Schrödinger software release 2021-4 [29]. All input molecules were defined as active using default settings. The values of the feature tolerance were 2 Å. All the default hypothesis settings were used except for the minimum number of features in the pharmacophore hypothesis, which was set at 3. The generation of conformers was performed by setting the target number of conformers to 50 within an energy range of 21 kJ/mol. The maximum number of generated hypotheses was set to 10. The minimum number of features in the pharmacophore hypothesis was fixed at 3, and at least 50% of the active molecules had to match the hypothesis.

For our study, we used hydrogen bond acceptors (A features), hydrophobic (H features), and aromatic rings (R features). The pharmacophores were generated from all molecular conformers. The algorithm selected a conformer for each molecule to align the molecules based on the common features shared by several molecules, which thus constituted a candidate pharmacophore [54]. A score was then calculated and assigned to each pharmacophore to determine the best pharmacophore [55,56].

## 5. Conclusions

The pharmacophore models generated from the components of the mixtures suggest that the RC mixture blending perception would not involve competition at the peripheral level while such competition could occur for the components of the masking mixture WL/IA.

The analysis of the elements of the clusters and the co-occurrences between odor notes points out the peculiar nature of the links between structure and odor notes of the molecules gathered in each cluster. It remains to be investigated how refining odor-structure relationships inside the clusters—considering the network of co-occurrences between the odor notes—could be a suitable way. Moreover, the models generated from the components of the aroma mixtures reveal in what way the pharmacophore approach provides significant additional information. Such approaches will constitute a powerful tool to explore odor-structure relationships.

## Figures and Tables

**Figure 1 molecules-28-04028-f001:**
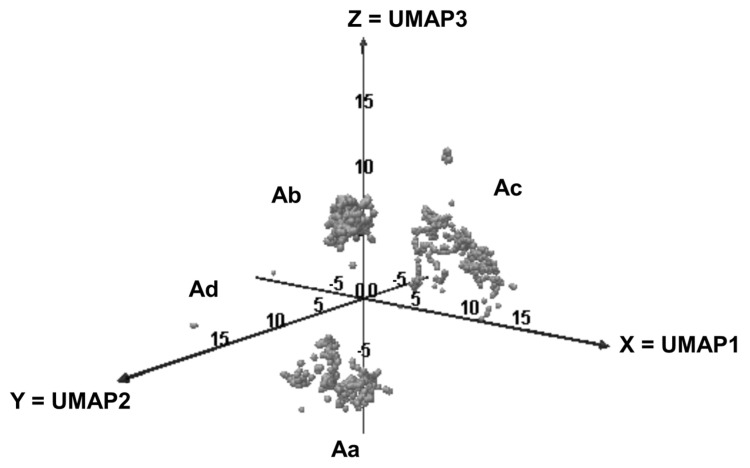
Distribution of the odorants in the 3D-UMAP space.

**Figure 2 molecules-28-04028-f002:**
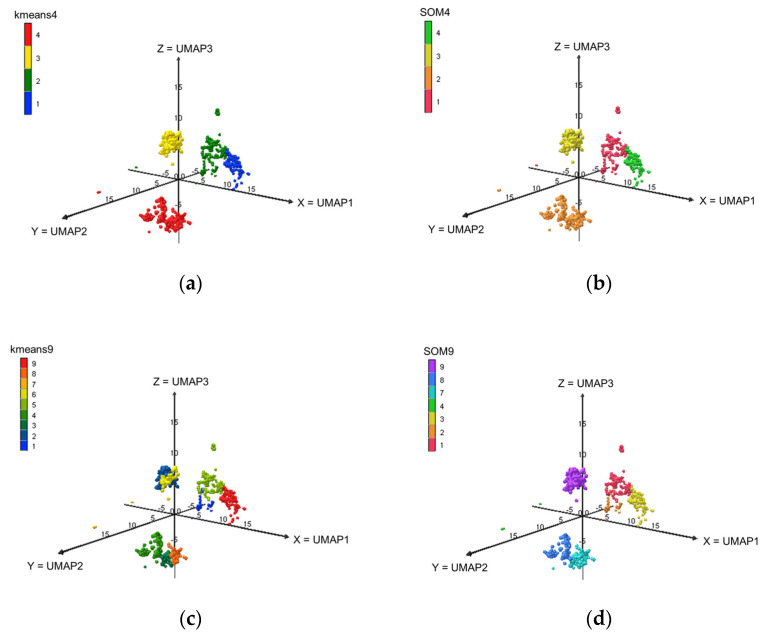
Visualizations of the elements of the dataset in the 3D-UMAP space based on the k-means and SOM clusters. (**a**) k-means4; (**b**) SOM4; (**c**) k-means9; (**d**) SOM9; (**e**) k-means16; (**f**) SOM16.

**Figure 3 molecules-28-04028-f003:**
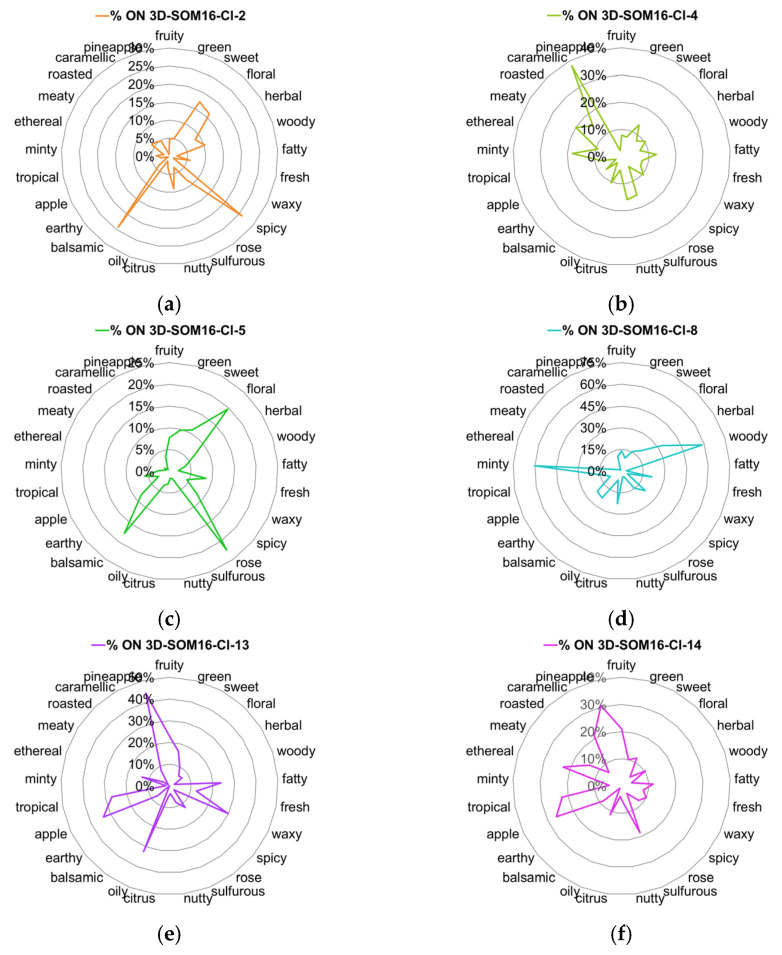
Radar charts of the %ON values for the 25 most frequent odor notes across clusters SOM16 that contain the molecules of interest. (**a**) SOM16-Cl-2, (**b**) SOM16-Cl-4, (**c**) SOM16-Cl-5, (**d**) SOM16-Cl-8, (**e**) SOM16-Cl-13, (**f**) SOM16-Cl-14.

**Figure 4 molecules-28-04028-f004:**
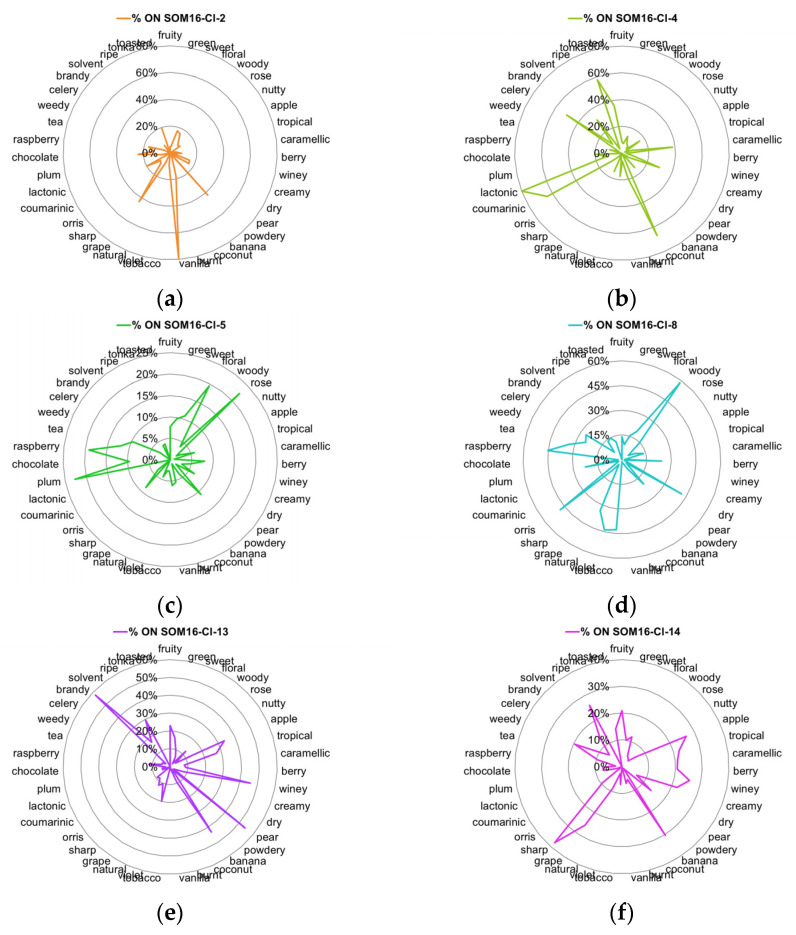
Radar charts of the %ON values for the 39 odor notes across clusters of the SOM16 clustering that contain the molecules of interest. (**a**) SOM16-Cl-2, (**b**) SOM16-Cl-4, (**c**) SOM16-Cl-5, (**d**) SOM16-Cl-8, (**e**) SOM16-Cl-13, (**f**) SOM16-Cl-14.

**Figure 5 molecules-28-04028-f005:**
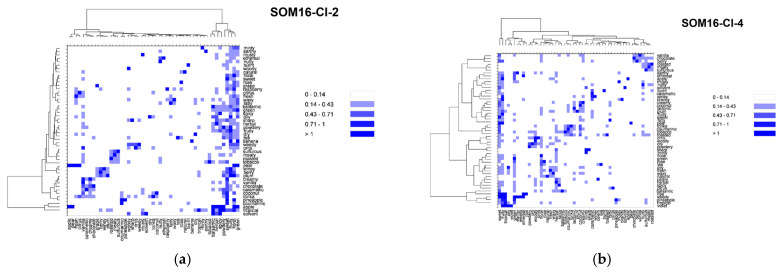
Heatmaps based on the nonsymmetrical square matrix of the odor notes relative co-occurrences of the 55 odor notes of the mixtures obtained from various groups of odorants. (**a**) SOM16-Cl-2; (**b**) SOM16-Cl-4; (**c**) SOM16-Cl-5; (**d**) SOM16-Cl-8; (**e**) SOM16-Cl-13; (**f**) SOM16-Cl-14; (**g**) all odorants of the database.

**Figure 6 molecules-28-04028-f006:**
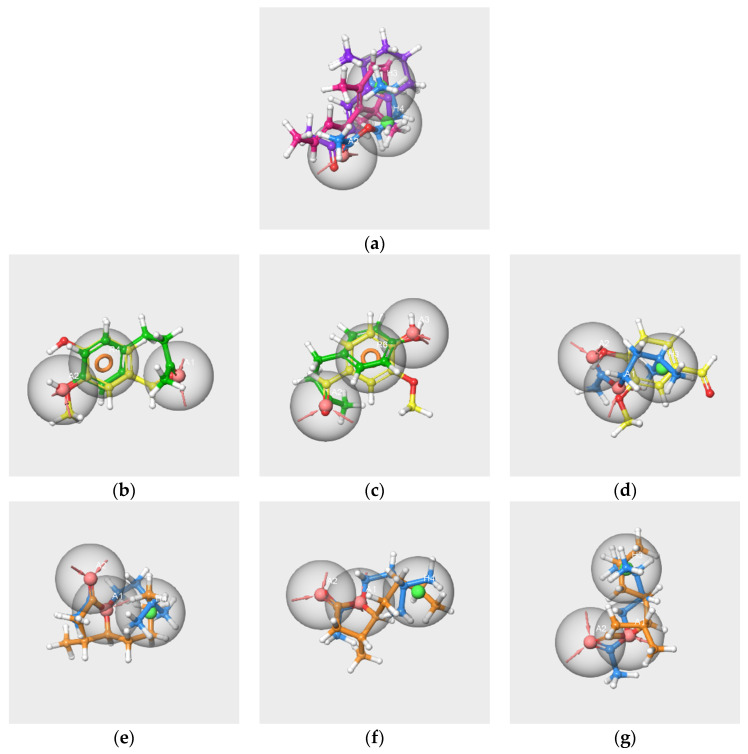
Pharmacophore hypotheses generated from the subsets V-IA-F-EA-bD-bI-s (**a**), V-IA-F-s (**b**–**d**) and WL/IA-s (**e**–**g**). V in yellow, IA in blue, F in green, bD in pink, bI in purple, WL in orange. The red spheres correspond to hydrogen bond acceptors, the green spheres correspond to hydrophobic groups, and the orange circles indicate aromatic rings R.

**Figure 7 molecules-28-04028-f007:**
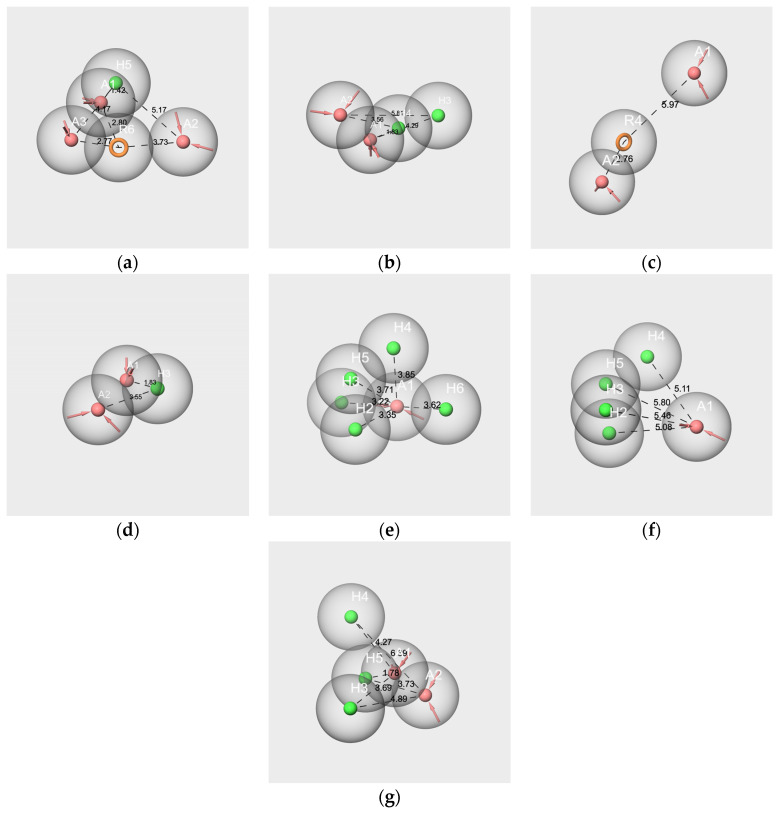
Pharmacophore best hypotheses generated from V-c (**a**), IA-c (**b**), F-c (**c**), EA-c (**d**), bD-c (**e**), bI-c (**f**), and WL-c (**g**). The red spheres correspond to the hydrogen bond acceptors. The green spheres correspond to the hydrophobic groups. Orange circles correspond to aromatic rings. The values indicate the distance in Ångström between two features. Only the distances between features A and H and A and R are represented.

**Figure 8 molecules-28-04028-f008:**
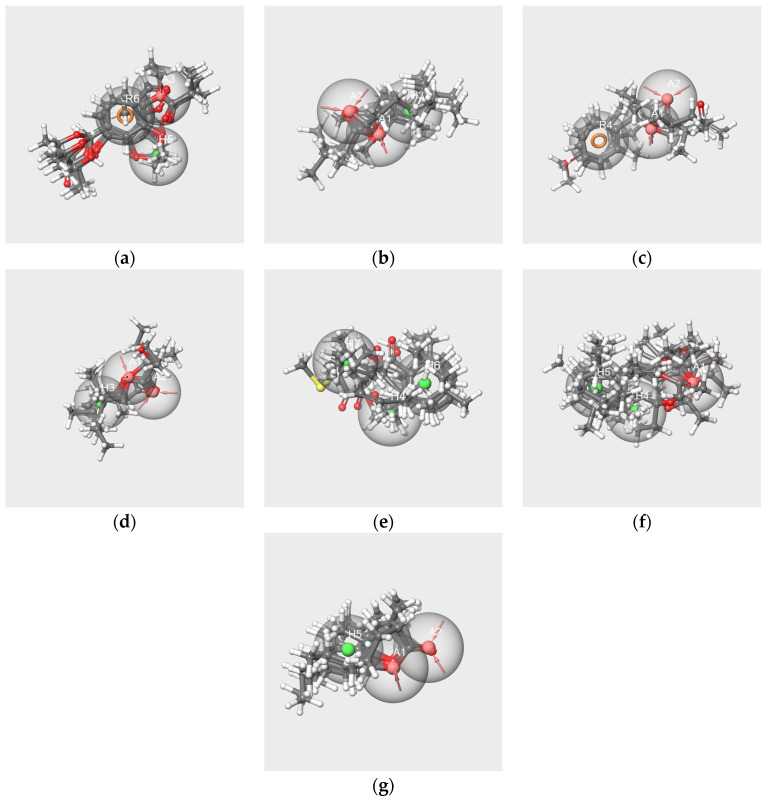
Best pharmacophore hypotheses obtained from the V-s (**a**), IA-s (**b**), F-s (**c**), EA-s (**d**), bD-s (**e**), bI-s (**f**) and W-s (**g**) subsets. The red spheres correspond to the hydrogen bond acceptors. The green spheres correspond to the hydrophobic groups. The orange circles correspond to aromatic rings.

**Figure 9 molecules-28-04028-f009:**
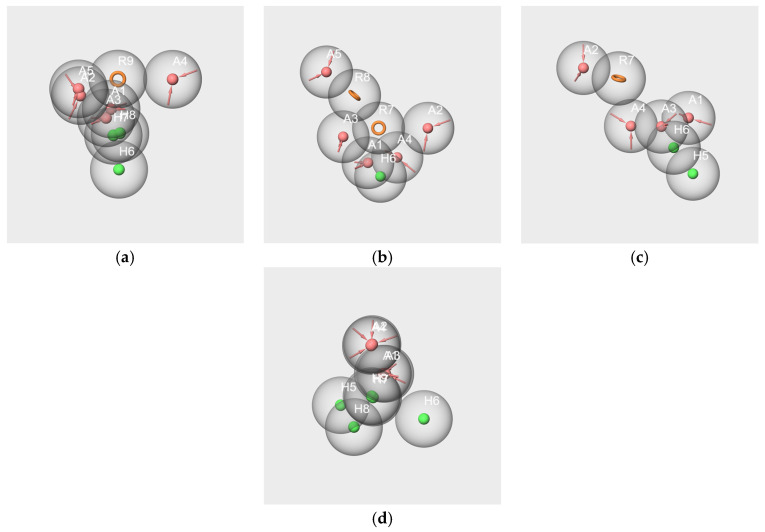
Pharmacophore mappings of hypotheses hyp-V-c and hyp-IA-c (**a**), hyp-V-c and hyp-F-c (**b**), hyp-IA-c and hyp-F-c (**c**), hyp-IA-c and hyp-WL-c (**d**).

**Figure 10 molecules-28-04028-f010:**
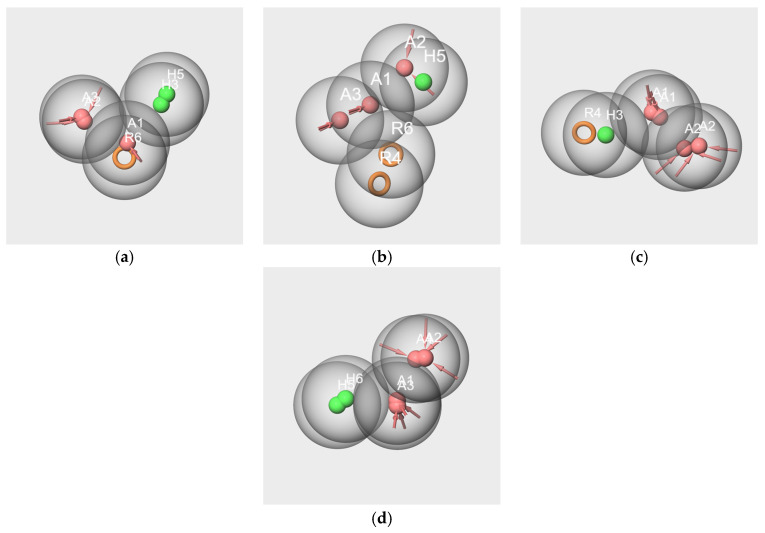
Pharmacophore mappings of the pharmacophore hypotheses hyp-V-s and hyp-IA-s (**a**), hyp-V-s and hyp-F-s (**b**), hyp-IA-s and hyp-F-s (**c**), hyp-WL-s and hyp-IA-s (**d**).

**Figure 11 molecules-28-04028-f011:**
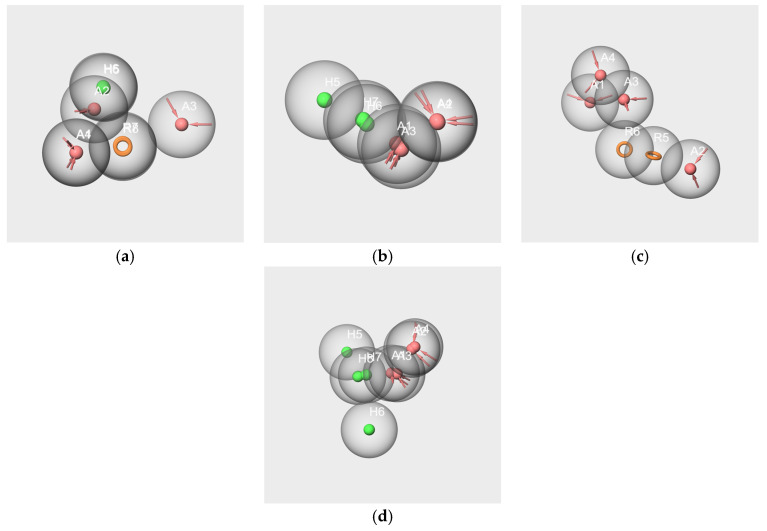
Pharmacophore mappings of the pharmacophore hypotheses hyp-V-c and hyp-V-s (**a**), hyp-IA-c and hyp-IA-s (**b**) and hyp-F-c and hyp-F-s (**c**), hyp-WL-c and hyp-WL-s (**d**).

**Table 1 molecules-28-04028-t001:** List of the components of the RC mixture and the WL/IA binary mixture and their respective odor notes.

Name	CAS	Odor Notes
Vanillin	121-33-5	Sweet; vanilla; creamy; chocolate
Isoamyl acetate ^1^	123-92-2	Sweet; fruity; banana; solvent; pear
Frambinone	5471-51-2	Sweet; berry; raspberry; ripe; floral; fruity
Ethyl acetate	141-78-6	Ethereal; fruity; sweet; weedy; green; sharp; brandy; winey
beta-Damascenone	23696-85-7	Fruity; floral; apple; plum; tea; rose; tobacco; natural; grape; raspberry; sweet
beta-Ionone	14901-07-6	Floral; woody; sweet; fruity; berry; tropical; violet; raspberry; dry; powdery orris
Whiskey lactone	39212-23-2	Tonka; coumarinic; coconut; toasted; nutty; celery; burnt; woody; lactonic; maple ^2^; lovage ^2^

^1^ Isoamyl acetate is common to the two mixtures. ^2^ The odor notes having less than 5 occurrences in the database are not considered in the analysis.

**Table 2 molecules-28-04028-t002:** Location of the molecules contained in the mixtures in the 3D-space areas and in the clusters.

Name	Area	k-means4	k-means9	k-means16	SOM4	SOM9	SOM16
Vanillin	Ab	3	2	8	3	9	2
Isoamyl acetate	Ac	1	9	4	4	3	13
Frambinone	Ab	3	2	8	3	9	5
Ethyl acetate	Ac	1	9	4	4	3	14
beta-Ionone	Aa	4	8	7	2	7	8
beta-Damascenone	Aa	4	8	7	2	7	8
Whiskey lactone	Aa	4	4	16	2	8	4

**Table 3 molecules-28-04028-t003:** List of molecules with an odor profile similar to those of the molecules of interest.

Cluster	Molecule of Interest	Molecule’s Subset with Similar Odor Profile
SOM16-Cl-2	Vanillin	Vanillyl isobutyrate; vanillin propylene glycol acetal; ethyl vanillin isobutyrate; 1-Ethoxy-2-methoxybenzene; ortho-dimethyl hydroquinone; ethyl vanillin; vanillyl acetate; vanillylidene acetone; vanillin hexylene glycol acetal; ethyl vanillin hexylene glycol acetal; ethyl vanillin propylene glycol acetal
SOM16-Cl-4	Whiskey lactone	7-Methyltetrahydronaphthalenone; delta-Heptalactone; Menthofurolactone; Octahydrocoumarin; Laitone; Coconut naphthalenone; (R)-tonka furanone; (+/−)-dihydromint lactone
SOM16-Cl-5	Frambinone	Anisyl isobutyrate; 4-hydroxyphenethyl alcohol; 4-(para-tolyl)-2-butanone; Tufurol acetate; 2-Methylbenzyl acetate; alpha-Methylbenzyl-propionate; Phenethyl-2-methylbutyrate; methyl 4-phenyl butyrate; benzyl acetoacetate
SOM16-Cl-8	beta-Ionone	beta-ionyl acetate; alpha-ionol; alpha-ionyl acetate; 3-Methylcyclohexyl acetate; beta-Irone; Campholene-acetate; Nopyl-acetate; 4-dimethyl ionone
SOM16-Cl-8	beta-Damascenone	plum damascone (high alpha); (Z)-alpha-damascone; Cyclohexylethyl isovalerate; Cyclohexylethyl valerate; 1-(3-(methyl thio)-butyryl)-2,6,6-trimethyl cyclohexene; 3-cyclohexene-1-carboxylic acid, 2,6,6-trimethyl-, methyl ester
SOM16-Cl-13	Isoamyl acetate	2-Methylbutyl-butyrate; hexyl acetate; isobutyl propionate; methyl butyrate; isopropyl propionate; methyl 4-methyl valerate; isoamyl butyrate; propyl acetate; butyl acetate; amyl acetate
SOM16-Cl-14	Ethyl acetate	2-Methylbut-2-enyl-formate; Isobutyl pyruvate; methyl acetate; methyl (E)-2-butenoate; ethyl 2-methyl butyrate; 2-methyl butyl propionate; isopropyl acetate; ethyl nitrite; hexyl lactate; methyl 3-hydroxybutyrate

**Table 4 molecules-28-04028-t004:** RMSD of the alignment of the pharmacophore hypotheses.

Pair of Hypotheses	RMSD
hyp-V-c and hyp-IA-c	0.5647
hyp-V-c and hyp-F-c	-
hyp-IA-c and hyp-F-c	-
hyp-WL-c and hyp-IA-c	0.1485
hyp-V-s and hyp-IA-s	0.5305
hyp-V-s and hyp-F-s	1.3647
hyp-IA-s and hyp-F-s	0.7449
hyp-WL-s and hyp-IA-s	0.3842
hyp-V-c and hyp-V-s	0.066
hyp-IA-c and hyp-IA-s	0.247
hyp-F-c and hyp-F-s	-
hyp-WL-c and hyp-WL-s	0.442

## Data Availability

Part of the data presented in this study are available in Appendix A. Full data are available on request from the corresponding author.

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
