# Peer review of "Combining the Classification and Pharmacophore Approaches to Understand Homogeneous Olfactory Perceptions at Peripheral Level: Focus on Two Aroma Mixtures"

_molecules, 2023, doi:10.3390/molecules28104028_

Round 1
Reviewer 1 Report
This manuscript focuses on the relationship between odor and the structure of odorants. An important defect is, the significant progress in this area was not addressed in this manuscript, such as Keller, Science, 2017, 355: 820-826; Ravia, Nature, 2020, 588: 118-123. So I think this paper needs major revision.
Author Response
Many odorant-odor notes models have been described recently. The progress in the area of relationship between odor and the structure of odorants are cited in the introduction (ref 11-18). The reference to the work of Keller et al. 2017 lacked and is now included (ref 11); however, the work of Ravia et al. appears to be in a different view and was no added.
In their publication, Keller et al. present the results obtained under crowd-sourced DREAM Olfaction Prediction Challenge. The purpose is to develop machine-learning algorithms to predict sensory characteristics (odor notes) of odorants. On the basis of the perceptual data from 338 molecules (and a test-set of the 69 molecules), the models have been capable of accurate prediction of pleasantness and intensity of the molecules, and about half of the 19 sensory characteristics (odor notes).
The study of Ravia et al. is based on a parallel between colors (wavelengths) and odors and aims to predict perceptual similarity, and the ensuing creation of olfactory metamers (“pairs of non-overlapping molecular compositions that generated identical odour percepts”). In that goal, they investigated the perceptual similarity estimates of 49,788 pairwise odorants, from 199 participants who smelled 242 different multicomponent odorants; 21 physicochemical features (sensory characteristics, odor notes) of the odorants were expressed into a single number (in radians). The obtained results indicate that it is possible to digitize the smell.
The purpose of our present take place in the same background of the understanding of links between olfactory perception and molecular structure of odorants, but not exactly by the same way. Nevertheless, our current goal is more to “understand” and “explore” than “predict”. We used a larger data set (more than 5000 odorants, 162 odor notes), and our approach do not involve sensory or psychophysical evaluation experiments; it is obviously unfeasible to test such a large number of molecules in a same study. That is why we used and compiled the information available in already existing data-bases. We did not use IA algorithms, our approach is first a SAR/QSAR study, the SAR/QSAR approaches enclosing use of dimensions reduction, classification and clustering methods, pharmacophore approaches belonging to SAR methods.
Note that two of us co-authored a recent publication (Achebouche et al. Sci Rep 2022, 12) that involves deep learning algorithms, Convolutional Neural Network (CNN) and Graphical Convolutional Network (GCN).
There are different but complementary approaches to lead to a better knowledge of olfactory perception. In that way, it seems not adequate to compare these works, each of which have their own advantages and limits.
Reviewer 2 Report
In their manuscript „Combining of classification and pharmacophore approaches to understand homogeneous olfactory perceptions: focus on two aroma mixtures“ (molecules-2331920), Rugard et al used a nonlinear dimensionality reduction technique (UMAP) together with clustering methods (k-means, SOM) to classify odors from a large dataset of odorants, as well as compounds from a 6-component configural/blending mixture (RC) and from a binary, masking mixture (WL/IA). They used subsets of these grouped molecules to generate pharmacophore models.
General remarks
The manuscript addresses a topical issue: understanding the olfactory perception of odorant mixtures. The paper is well written, with precise referencing, and is easy to follow. The present work, however, appears as a sequel of the authors‘ previous work, this time, however, adding pharmacophore modeling. The title is rather descriptive, though, and the ‘take-home-message‘ (see Abstract and end of Discussion) appears to be „stay tuned“. Maybe a more informative title could be built around the message of the next-to-last sentence in the abstract „...pharmacophore models suggest that WL and IA could have common binding site(s) at the peripheral level...“?
1. Why did the authors compare a 6-component configural mixture with a binary mixture when there are food-related configural binary mixtures? Are the mixtures in the present study actually based upon food-relevant, quantitation-based odor activity values, or are they artificially set to be iso-intense?
2. While the pharmacophore hypotheses raised in this study have been convincingly tested via alignments etc., it would have been advantageous to test the respective pharmacophore hypothesis with human odorant receptors in in-vitro functional tests. Also, docking studies come to mind, using known selective, cognate odorant/receptor pairings, at least for β-ionone/OR5A1 (S.R. Jaeger, et al., A Mendelian trait for olfactory sensitivity affects odor experience and food selection, Curr. Biol. 23 (16) (2013) 1601–1605.), or vanillin/OR10G3 (J.D. Mainland, et al., The missense of smell: functional variability in the human odorant receptor repertoire, Nat. Neurosci. 17 (1) (2014) 114–120), and related odorants.
In general, it will be most informative to identify the most sensitive and/or selective responding odorant receptors for the odorants under investigation - the authors rightfully point to their ongoing in vitro functional tests.
Major points
Table 1: The FEMA Flavor Profile of whiskey lactone is „Butter, Cocoa, Coconut“, however, lacking a „woody“ descriptor. Did the authors check the odor quality themselves? Was the odor quality „woody“ by any means concentration-dependent? Were the odorants in the RC mixture adjusted to be iso-intense?
Line 162ff With regard to the odor notes (labels) of their dataset and the UMAP-derived clustering/classification: Did the authors took into account odor label correlation, odor label dependency, or label imbalance?
Lines 23, 24, 591, 602ff From their pharmcophore modeling, the authors infer that „the key components of RC mixture do not share a common binding site or binding mode.“, see Abstract, lines 23, 24: „...that would be excluded for the components of RC...“, and interprete this as follows: „...RC mixture blending perception does not originate at the peripheral level...“. I strongly object this notion for two reasons:
1. The authors disregard e.g. allosterism. There are plenty of papers and reviews pointing to manifold pharmacodynamic principles governing the interaction of odorants at the receptor level (e.g. S. M. Kurian et al., Cell Tissue Res 2021).
2. How do the authors envision broadly tuned receptors (e.g. OR2W1, see F. Haag et al., Food Chemistry 2022)?
Moreover, all this can be tested in in-vitro functional studies. The authors‘ discussion needs to be more careful at this point.
Minor points
Table 1. The CAS information for vanillin needs to be updated - in their manuscript the authors give a deprecated CAS registry number.
Line 233 „Error! Reference source not found“.
Line 588 „...has a specific structure its own distinct.“ What does this mean? Does this sentence lack something?
Author Response
Answer to general remarks
The work presented in the manuscript is indeed an enlargement of the approach previously published by refining the distribution of odorants according structure-odor groups.
We have considered the following title “Combining of classification and pharmacophore approaches to understand homogeneous olfactory perceptions at peripheral level: focus on two aroma mixtures”.
However, it seemed hazardous to evoke in the title the role of the peripheral level only on the basis of the obtained pharmacophores, that is to say without proposal of a biological mechanism based on experiments. Indeed, we only suggest the possibility of an interaction at a same binding site for WL and IA, which could be result in a competitive binding of the two ligands. Nevertheless, at this step and on the basis of the present results, we do not provide an answer to assess or discard this hypothesis. The present work is a part of a broader project that includes in vitro, ex vivo and in vivo experiments, and whose results will be soon submitted for publication. We aim so to answer to some issues addressed in this manuscript.
Answer to point 1.
There is not a direct comparison of the intensities between a configural mixture and a masking mixture but a focus on two cases of homogeneous perception previously studied. RC and WL/IA mixtures are food odors, but were not extracted from specific foods or beverages. They were selected because they have been widely studied by the research teams of CSGA since several years by psychophysical experiments and sensory evaluations.
The 6-components mixture have been designed in the context of previous works (Le Berre et al., 2008 Chem. Senses 33: 193-199 DOI 10.1093/chemse/bjm080); Sinding et al. 2013 Plos One 8 e53534 DOI 10.1371/journal.pone.0053534). Preliminary check for iso-intensity between odor stimuli have been carried out, and the ratio leading to a configural perception of has been especially studied, as well in human as in newborn rabbits. Besides, the study of the masking WL/IA was based on the study of woody/fruity notes known to contribute to some of the major notes in wines, so related on a real beverage (Atanasova et al. 2005 Chemical Senses 30, 209-217 DOI 10.1093/chemse/bji016; Chaput et al., 2012 Eur. J. Neurosci. 35: 584 DOI 10.1111/j.1460-9568.2011.07976.x). In these studies, previously carried out by the research team in Dijon, the ratios have been studied using several possible combinations. Nevertheless, neither the conditions of the experiments nor ratios or odor intensities did not need to be considered in order to perform our present computational study. The reference Chaput et al. 2012 was added (Ref. 26).
Actually, we considered the intrinsic molecular properties in relation with the odor notes regardless of perceived intensities (as it is for binary data: 1 when the odor note appears in the odor description, 0 otherwise). The used structural properties (fingerprints) are intrinsic properties of the molecules regardless of their concentrations in vapor phase and their perceived intensities.
Obviously, the ratios between the component must be considered for in vitro studies, as well the intensities for in vivo studies that exceed the context of the present work and consequently are not reported in this manuscript.
Answer to point 2.
We are thankful to you for your comments, and we agree with your advices. In-vitro functional tests with odorant receptors have been recently carried out in this view, and the results will be soon submitted for publication. Moreover, in vivo experiments in human are also planned. We agree that docking studies would be very suitable, and it is hoped that they will be undertaken.
Answer to major points
Table 1
As regards of the “woody” note of WL, it is admittedly possible that it is related to the concentration of the molecules. We pointed out this note because our work is based on the previous studies related to the "woody" note carried by WL in wine (Atanasova et al. 2005). Nevertheless, “woody” do not appears as characteristic of WL odor in most reported odorant descriptions whereas “creamy”, “coconut” and “coumarinic” are consensual notes for the odor of this molecule. Besides, this description is reported in Flavor Base: “Mosciano, et. al. (1997) describe the odor as "Coumarinic, coconut, lactonic, woody, maple and lovage with a slightly toasted, nutty nuance" and the taste (at 0.5 ppm) as "Woody, coumarinic, coconut, lactonic, creamy and nutty with a toasted nuance". Suggested Applications - Coumarin, coconut, whiskey, woody and casky notes. (Perfumer & Flavorist, Vol. 22, March/Apnl 1997, pp. 70-71)”
In fact, it seems that WL carries a complex odor; the identification of their targets ORs currently performed in our research group could provide a mean for the explanation of this complexity.
The odor qualities weren’t checked by us; we used the odor descriptions reported in the databases Flavor Base and the Good Scents Company. Nevertheless, the odor quality has been checked in the context of the initial works (Atanasova et al. 2005 Chemical Senses 30, 209-217 DOI 10.1093/chemse/bji016; Le Berre et al., 2008 Chem. Senses 33: 193-199 DOI 10.1093/chemse/bjm080); Sinding et al. 2013 Plos One 8 e53534 DOI 10.1371/journal.pone.0053534); no later sensory study has been performed by the authors of the present work (our purpose is to establish a computational structure-odor relationships approach, and do not involve sensory evaluations).
In aqueous solution, the fruity note perception is enhanced by low concentrations of WL while there is a perceptual a dominance of the woody note for iso-intense mixtures WL/IA; the woody odor intensity increases with the WL concentration (Le Berre et al., 2007 Food. Qual. Prefer. 18: 901 DOI 10.1016/j.foodqual.2007.02.004; Chaput et al. 2012 Eur. J. Neurosci. 35: 584 DOI).
The configural perception (blending effect) the RC mixture has been compared to its components, iso-intensity being checked for each odor set (single component or mixture); nevertheless, the combination of specific odorants is not sufficient. Indeed, the processing that induce the configural processing of the RC mixture depends on the relative proportions of these components. Therefore, the components of the RC mixture would be not iso-intense if tested individually at the same concentration than used in the blending mixture (Sinding et al., 2013).
Line 162
The odor notes were taken into account neither for UMAP nor clustering calculations. We used only the molecular properties encoded by fingerprints to perform the calculations. The odor notes are afterwards used to analyze the relative frequencies of the odor notes (most frequent and/or carried by the components of the mixtures) in each cluster against the whole data set (%ON); we also considered the weight of the molecules carrying a these notes in each cluster (%OM).
Lines 23, 24, 591, 602f
Answer to point 1 (allosterism)
We do not disregard allosterism, nor the possibility that at least two sub-sites exist in a large cavity. “Common binding site or binding mode” do not means “common olfactory receptor”. Proteins receptors and globular proteins can have several binding sites at different regions of their structure.
Nevertheless, when no common alignment exists for several molecules, it is very unlikely (the pharmacophores are hypotheses, not the absolute reality) that they can bind exactly to the place at any receptor. In other words, the ligands do not bind the same residues or same amino acid chains of the receptor. Besides, two molecules can bind to the same receptor according two different regions or modes of binding. It can be the case of the components of the RC mixture, and also those of WL and IA (even they have, additionally, at least a common binding site for which they can have competitive affinities). At this step and on the basis of the present results, it is no possible to resolve this point that should be clarified by the in vitro functional experiments on olfactory receptors.
We completed the discussion in that way and added several refences to develop the cases of the existence of several binding sites at a same protein.
Answer to point 2: broadly tuned receptors
The broadly tuned receptors have several binding sites (at different zones of the receptor and/or in a large cavity (Sanz et al. 2008 Chem. Senses 33: 639 DOI 10.1093/chemse/bjn032; Belhassan et al. 2017, 18, 257 DOI 10.1016/j.ejenta.2017.11.004). According to the previous answer, it is not excluded that some components of RC mixture can bind to a same OR, but not at the same small cavity.
A study using 3-D QSAR pharmacophore approach was carried out by one of us about fifteen years ago (Sanz et al. Chem. Senses 2008, 33, 639). This work allowed to identify two alignments geometries of ligands of OR1G1 on the two generated pharmacophore models. This result was successfully used for further molecular docking experiments (Charlier et al. 2012 Cell. Mol. Life Sci. 69: 4205 DOI 10.1007/s00018-012-1116-0 ; Launay et al. 2012 Protein Eng. Des. Sel. 25: 377 DOI 10.1093/protein/gzs037). Nevertheless, 3D-QSAR approach requests the use of quantitative affinity data of a sufficiently large set of ligands (several dozen) for a receptor obtained in the same laboratory, which are rarely available until now. The data provided in the recent study of Hagg and al. 2022 will allow such pharmacophores generations on the basis of the EC50 values.
Some globular proteins are also known to possess several binging sites. It is the case of beta-lactoglobulin (Lübke et al. J. Agric. Food Chem. 2002, 50, 7094; Tavel et al. Food Chem. 2010, 119, 1550); two pharmacophores and ligands alignments related to each binding site had been proposed (Tromelin and Guichard J. Agric. Food Chem. 2003, 51, 1977; Tromelin and Guichard Flavour Frag. J. 2006, 21, 13).
We added references the following references to illustrate the cases of multi-site and broadly tuned receptors:
- Sanz, G.; Thomas-Danguin, T.; Hamdani, E. H.; Le Poupon, C.; Briand, L.; Pernollet, J. C.; Guichard, E.; Tromelin, A., Relationships Between Molecular Structure and Perceived Odor Quality of Ligands for a Human Olfactory Receptor. Chem. Senses 2008, 33, 639-653. DOI 10.1093/chemse/bjn032.
- Launay, G.; Teletchea, S.; Wade, F.; Pajot-Augy, E.; Gibrat, J. F.; Sanz, G., Automatic modeling of mammalian olfactory receptors and docking of odorants. Protein Eng. Des. Sel. 2012, 25, 377-386. DOI 10.1093/protein/gzs037.
- Charlier, L.; Topin, J.; Ronin, C.; Kim, S. K.; Goddard, W. A.; Efremov, R.; Golebiowski, J., How broadly tuned olfactory receptors equally recognize their agonists. Human OR1G1 as a test case. Cell. Mol. Life Sci. 2012, 69, 4205-4213. DOI 10.1007/s00018-012-1116-0.
- Kim, S. K.; Goddard, W. A., Predicted 3D structures of olfactory receptors with details of odorant binding to OR1G1. J Comput Aided Mol Des 2014, 28, 1175-1190. DOI 10.1007/s10822-014-9793-4.
- Belhassan, A.; Zaki, H.; Chtita, S.; Benlyas, M.; Lakhlifi, T.; Bouachrine, M., Study of interactions between odorant molecules and the hOR1G1 olfactory receptor by molecular modeling. Egyptian Journal of Ear, Nose, Throat and Allied Sciences 2017, 18, 257-265. DOI 10.1016/j.ejenta.2017.11.004.
- Fukutani, Y.; Abe, M.; Saito, H.; Eguchi, R.; Tazawa, T.; de March, C. A.; Yohda, M.; Matsunami, H., Antagonistic interactions between odorants alter human odor perception. bioRxiv 2022, 2022.08.02.502184. DOI 10.1101/2022.08.02.502184.
- Zak, J. D.; Reddy, G.; Vergassola, M.; Murthy, V. N., Antagonistic odor interactions in olfactory sensory neurons are widespread in freely breathing mice. Nat. Commun. 2020, 11. DOI 10.1038/s41467-020-17124-5.
- Inagaki, S.; Iwata, R.; Iwamoto, M.; Imai, T., Widespread Inhibition, Antagonism, and Synergy in Mouse Olfactory Sensory Neurons In Vivo. Cell Reports 2020, 31, 107814. DOI 10.1016/j.celrep.2020.107814.
- Xu, L.; Li, W. Z.; Voleti, V.; Zou, D. J.; Hillman, E. M. C.; Firestein, S., Widespread receptor-driven modulation in peripheral olfactory coding. Science 2020, 368, eaaz5390. DOI 10.1126/science.aaz5390.
- de March, C. A.; Titlow, W. B.; Sengoku, T.; Breheny, P.; Matsunami, H.; McClintock, T. S., Modulation of the combinatorial code of odorant receptor response patterns in odorant mixtures. Mol. Cell. Neurosci. 2020, 104, Unsp 103469. DOI 10.1016/j.mcn.2020.103469.
Answers to minor points
Table 1. CAS information for vanillin: The CAS number was corrected (121-33-5)
Line 233 „Error! Reference source not found“. The mistake line 233 was corrected, as well as a similar mistake line 637.
Line 588 “has a specific structure its own distinct.“ The sentence was modified as follow: “That means that each key component of the RC mixture has a specific spatial structure that differs from those of the other components of this mixture.”
Round 2
Reviewer 2 Report
No further comments.